# The Impact of Family Wealth on Asset Return: A Moderated Chain Median Model Partially Explaining Wealth Inequality

**DOI:** 10.3390/bs14111048

**Published:** 2024-11-06

**Authors:** Tianye Tu

**Affiliations:** Department of Business, Gachon University, Seongnam 13120, Republic of Korea; tutianye2018@yeah.net

**Keywords:** wealth inequality, wealth return heterogeneity, COR theory, risk preference, asset-holding period, disposition effect

## Abstract

Global wealth is distributed highly unequally, and this issue has worsened in recent years. Such inequality threatens human well-being and fundamental values. In response, this paper aims to explore the wealth inequality phenomenon from the perspective of investment psychology. Recognizing wealth return heterogeneity as a crucial source of inequality, the paper establishes a psychological model focused on two investment skill factors that can partially explain this heterogeneity. The theoretical foundation of this model includes the conservation of resources (COR) theory, prospect theory, and performance determinants theory. In our model, risk preference and asset-holding periods sequentially mediate the relationship between family wealth and asset returns. Moreover, risk preference and the disposition effect are identified as serial mediators in this relationship. Financial literacy also moderates the relationship between the risk preference, asset-holding period, and disposition effect. This proposed model not only provides a richer explanation for wealth return heterogeneity through the lens of investment skill but also extends the application of COR theory to investment psychology, thereby enhancing our understanding of resources. Moreover, it offers a novel explanation for the relationship between risk preference and the deposition effect, enriching prospect theory. Finally, the inclusion of financial literacy’s role broadens the scope of performance determinants theory.

## 1. Introduction

Global wealth distribution has become increasingly unequal, a trend that has intensified over recent years [1]. With the inclusion of the United States, Europe, and China, the wealth share of the top 1% grew from 28% in 1980 to 33% today [1]. In contrast, the bottom 75%’s wealth share has remained around 10% from 1980 to the present [1]. This significant wealth inequality can lead to a range of severe social problems, including opportunity disparities for future generations, unequal political representation, and detrimental effects on economic growth [2]. These problems threaten the fundamental values and well-being of humanity [2]. Consequently, clarifying the mechanism behind wealth inequality, which is critical for solving this problem, has become a key focus for both practitioners and scholars [2,3,4].

Originally, many scholars believed that income inequality was the main determinant of wealth inequality [4]. However, this theoretical perspective cannot fully explain why wealth distribution is far more unequal than income distribution [4]. Therefore, scholars have begun to explore other factors that contribute to wealth inequality, with wealth return heterogeneity receiving increasing attention. This factor has been recognized as an important source of inequality by many researchers [4,5,6]. There are two main explanations for wealth return heterogeneity. First, wealthy families often have a higher risk preference, leading them to choose asset portfolios that are riskier but profitable [4,5,6]. Second, even when investing in the same type of assets, wealthy families receive higher returns due to superior investment skills [6]. Both explanations are supported by empirical evidence [5,6]. However, research on the investment skill explanation is scarce, and it remains unclear which specific investment skills contribute to wealth return heterogeneity [6]. As a result, this paper aims to theoretically explore the investment skill factors that contribute to wealth return heterogeneity, especially focusing on the role of the asset-holding period and disposition effect.

There is scant research exploring the relationship between family wealth levels and asset-holding periods. As a result, this research aims to theoretically explore this relationship from the perspective of risk. To elucidate this relationship, risk preference will be introduced as a mediator. Additionally, there is empirical evidence supporting a negative relationship between wealth and the disposition effect [7]. However, the role of risk preference in this relationship remains unclear. Therefore, this paper aims to theoretically investigate the mediating role of risk preference.

Moreover, this paper also plans to explore the moderating role of financial literacy in the relationship between risk preference, asset-holding periods, and the disposition effect. Here are two reasons for which this research objective is valuable. First, although there are many studies supporting the positive impact of financial literacy on good financial behavior or practice such as saving, retirement planning, investment participation, cash management, credit management, and risk management [8,9,10,11,12], little research concerns the impact of financial literacy on the two financial behaviors (asset-holding period and disposition effect). Second, many scholars have confirmed the positive relationship between financial literacy and wealth accumulation [9,13,14], but little is known about the specific mechanism of the impact of financial literacy on wealth accumulation. It is helpful for revealing the mechanism to explore the impact of financial literacy on asset-holding periods and the disposition effect that are linked with wealth accumulation.

The primary research method employed in this research is a theoretical analysis. This choice is supported by several reasons. First, the research topic—investment skill factors contributing to wealth return heterogeneity—is relatively novel [4], and a theoretical framework is yet to be established, necessitating a theoretical explanation of this topic, as it can offer guidance for subsequent empirical research. Second, obtaining the data necessary to confirm the model proposed in this paper poses a significant challenge for ordinary researchers, especially when it involves gathering data on asset-holding periods, financial literacy, and asset return simultaneously. Third, the main aim of this paper is to offer new theoretical insights into this emerging topic, based on well-documented current theories, rather than focusing on empirical validation.

In the final part of this introduction, the author summarizes the objectives and primary contribution of this study. First, this study aims to explore the impact of two investment skill factors (asset-holding period and disposition effect) on wealth return heterogeneity, which enriches the current explanation for wealth return heterogeneity. Second, this study plans to explore the mediating role of risk preference in the relationship between the family wealth level, asset-holding period, and disposition effect, which is scarce in the current literature. This exploration can enhance our understanding of the above relationship. Third, this study wants to discuss the moderating role of financial literacy in the relationship between the risk preference, asset-holding period, and disposition effect. This can extend the research about the effect of financial literacy and enrich the explanation for the positive relationship between financial literacy and wealth accumulation.

## 2. Literature Review and Propositions

### 2.1. Applying Conservation of Resources (COR) Theory to Explanation of Relationship Between Family Wealth, Risk Preference, Asset-Holding Period, and Asset Return

In this part, the author will utilize COR theory to explain the serial mediating role of risk preference and asset holding in the relationship between family wealth and asset return. Currently, COR theory is one of the most influential theories in organizational psychology [15]. The fundamental premise of COR theory is that individuals strive to obtain and conserve resources for survival, a phenomenon that is central to human behavioral genetics [15]. Additionally, the theory posits that individuals must invest resources to prevent resource loss and to acquire more resources [15]. The specific content of this theory describes the principles governing how individuals obtain and conserve resources. Although primarily applied in organizational psychology, COR theory is also relevant to the topic of this paper, which belongs to investment psychology. This relevance stems from the theory’s potential to explain why individuals with initially rich financial resources can ultimately accrue more financial resources than less advantaged individuals through investment. As a result, this paper will apply COR theory to elucidate the relationship between the family wealth, risk preference, asset-holding period, and asset return, thereby clarifying the mechanisms behind wealth return heterogeneity.

In this study, the author adopts the definition of family wealth as outlined by Pfeffer and Schoni (2016) [2]. Family wealth is described as the net worth, which is the sum of all family assets minus all family liabilities. Family assets include financial assets such as transaction accounts, stocks, bonds, and retirement accounts, as well as non-financial assets such as residential properties, vehicles, non-residential properties, and business equity. Family liabilities include both short-term debt, like credit card debt, and long-term debt, comprising mortgages, car loans, and student loans. According to COR theory, a resource is defined as anything that people perceive as helpful in achieving their goals [16]. Given that wealth fundamentally satisfies people’s needs for consumption and social status, family wealth is considered a valuable resource within the COR theory framework. Moreover, Hobfoll et al. (1989), the originators of the COR theory, identified socioeconomic status as a crucial resource within the COR framework [17]. Since family wealth is intricately intertwined with socioeconomic status, the notion that family wealth constitutes an important resource in COR theory is well supported.

Risk preference refers to the degree to which an individual prefers to engage in activities or behaviors that exhibit significant variation in returns, without knowing whether they finally result in losses or gains [18]. Risk preference is an important psychological resource because it is positively associated with investment returns (wealth) [5,6]. Individuals with more family wealth exhibit higher risk preferences because they can afford the potential losses of taking risks [19]. In contrast, for individuals with a low level of family wealth, their ability to undertake potential loss of taking risks is weak. Moreover, as wealthy individuals usually have more financial resources for investment, they are more likely to have higher expectancy about the asset return. This higher expectancy motivates them to undertake more risk and pursue higher asset return. The positive relationship between family wealth and risk preference confirms the basic view of COR theory that individuals with more resources are more capable of gaining additional resources compared to those with fewer resources [15]. This viewpoint is also supported by empirical studies [5,6].

In this paper, the asset-holding period is defined as the length of time an investor retains an asset before its sale. In modern finance, scholars usually measure the risk of an investment using return variation [20]. Empirical evidence shows that a longer asset-holding period is associated with more variation in asset returns [20], indicating that individuals who hold an asset for extended periods undertake higher risks. Bach et al. (2019) and Fagereng et al. (2020) also agree with this idea. In their articles, they point out that the asset-holding period can reflect the risk undertaken by the investor [5,6]. Moreover, holding an asset for a longer period not only means higher risk of price variation but also increases liquidity risk of investors. As previously mentioned, individuals with more family wealth exhibit higher risk preferences, leading them to assume higher risks and hold assets for longer durations. This paper also examines the relationship between family wealth and asset-holding periods through the lens of COR theory. According to COR theory, individuals with more resources are more proactive in resource investment behavior [21]. Holding an asset for a longer period can be considered an active resource investment behavior, as it consumes more time and resources while entailing higher investment risks. Moreover, holding an asset for a longer duration is associated with higher return [20]. Consequently, based on COR theory, this paper predicts a positive relationship between the family wealth and asset-holding period.

The basic principle of investment states that return is positively related to the risk undertaken by investors [22]. As previously mentioned, individuals who hold an asset for a longer period assume higher risks, and thus, it is reasonable to expect that they would achieve higher returns. Additionally, the long-term price trend of main assets generally moves upwards [22], supporting the idea that longer asset-holding periods are associated with higher returns. Empirical evidence also validates this concept [20]. This paper further examines the relationship between the asset-holding period and return from the perspective of COR theory. According to COR theory, individuals must invest resources to obtain resources [18]. Since a longer asset-holding period can be viewed as active resource investment behavior, it is logical to associate this behavior with higher returns.

Considering the related empirical evidence and the prediction of COR theory, this research proposes the following:

*P1:* 
*Individuals with more family wealth exhibit higher risk preferences.*


*P2:* 
*Individuals with more family wealth maintain a longer asset-holding period, regardless of the type of asset.*


*P3:* 
*Individuals with more family wealth achieve higher asset returns, regardless of the type of asset.*


*P4:* 
*The risk preference and asset-holding period serially mediate the relationship between family wealth and asset return.*


### 2.2. Family Wealth and Disposition Effect

The deposition effect is one of the most widely observed biases in the behavior of individual investors [23]. It refers to the tendency of an investor to sell assets that have increased in value since purchase and hold onto the assets that have decreased in value [24,25,26]. This effect is observed across various asset markets such as the stock market [24], derivative product market [25], real estate market [26], and bond market [27]. Empirical evidence sheds light on a negative relationship between household income and the disposition effect [7]. Given that household income embodies a dynamic flow of family wealth, this research posits that family wealth demonstrates a relationship akin to the disposition effect. Previously, scholars have elaborated on this phenomenon by proposing that wealthier investors benefit from superior access to investment advice from professional agents and are more adept at processing information; as a result of which, they are more likely to circumvent the disposition effect [7]. As a result, this study proposes the following:

*P5:* 
*Individual investors’ family wealth is negatively related to the disposition effect.*


### 2.3. Risk Preference and Disposition Effect: Prospect Theory Perspective

In contrast to the explanation provided by Dhar and Zhu (2009) [7], which was mentioned before, this paper attributes the negative relationship between family wealth and the disposition effect to risk preference. Having discussed the positive relationship between family wealth and risk preference, this paper now plunges into the negative relationship between risk preference and the disposition effect through the lens of prospect theory.

Prospect theory is one of the most influential theories in behavioral economics. Here are some fundamental aspects of prospect theory that have been validated by many behavioral experiments [28]. First, the theory proposes that people assess utility based on gains and losses, which are measured by comparison with a reference point, rather than on the absolute amount of wealth [28]. People are more sensitive to losses than to gains of the same magnitude, a phenomenon known as loss aversion [28]. Third, as the magnitude of gains or losses increases, people’s sensitivity to these gains or losses diminishes. This effect results in a value function that is convex in the area of losses and concave in the area of gains. Fourth, individuals exhibit risk aversion when they gain, along with risk-seeking behavior when they face losses.

The original value function of prospect theory is defined as $\left\{v\left(x\right)=x^{\alpha },x\geq 0,0<\alpha \leq 1;v(x)=-\beta {\left(-x\right)}^{\alpha },x<0,\beta >1,0<\alpha \leq 1\right\}$. In this function, x represents gains or losses relative to the reference point [28]. This function reflects the four basic views of prospect theory previously mentioned [28]. People are more sensitive to losses than to gains of the same magnitude, so β > 1. Moreover, the reason for 0 < α ≤ 1 is that as the magnitude of gains or losses increases, people’s sensitivity to these gains or losses diminishes. According to mathematical knowledge, when 0 < α ≤ 1, the curve gradually becomes gentle with gains or losses increasing, which can reflect the diminished sensitivity to gains or losses. Next, this paper will discuss the form of the value function for individuals with low- and high-risk preferences. According to the previously mentioned definition of risk preference, individuals with high-risk preferences pursue higher gains and can accept higher losses compared to those with low-risk preferences. As a result, the main difference between these two types of individuals is that those with low-risk preferences are more sensitive to gains or losses. Specifically speaking, compared with high-risk-preference individuals, individuals with low-risk preference become more satisfied when facing gains and become more dissatisfied when facing losses as the expected range of return of these individuals (low-risk preference) is relatively narrow. Based on this primary distinction, this paper constructs the value functions for individuals with high- and low-risk preferences.

This research describes the value curve of individuals with high-risk preferences using the original value function, represented as $\left\{v\left(x\right)=x^{\alpha },x\geq 0,0<\alpha \leq 1;v(x)=-\beta {\left(-x\right)}^{\alpha },x<0,\beta >1,0<\alpha \leq 1\right\}$. For individuals with low-risk preferences, their value function is modified to $\left\{v\left(x\right)=\rho x^{\alpha },x\geq 0,0<\alpha \leq 1,\rho >1;v(x)=-\gamma \beta {\left(-x\right)}^{\alpha },x<0,\beta >1,0<\alpha \leq 1,\gamma >1\right\}$. As individuals with low-risk preferences are more sensitive to gains and losses than individuals with high-risk preferences, ρ,γ>1. Figure 1 shows the comparison of the function curves for these two types of investors.

Based on the differences in the value curves, this research can infer that individuals with low-risk preferences tend to experience greater satisfaction and have a stronger inclination to secure the gain when confronted with a gain on a particular asset. This heightened desire to realize gains translates into a higher propensity to sell the asset to crystallize the profit. Conversely, when facing a loss on a particular asset, individuals with low-risk preferences undergo elevated dissatisfaction. This dissatisfaction fuels a strong desire to recover the lost value, leading them to hold onto the asset in anticipation of a future price increase. So, they are more likely to hold the asset in anticipation to wait for the rising of the price of the asset. As a consequence, risk preference is negatively related to the disposition effect. This theoretical analysis and its result are consistent with prospect theory’s fundamental premise that gains prompt people to avoid risk, while losses encourage risk taking. This analysis result is supported by the study of Andrikogiannopoulou and Papakonstantinou (2020) that is a simulation study [29]. The simulation result shows that low-risk preference enhances the disposition effect. As a result, this paper proposes the following:

*P6:* 
*An individual’s risk preference is negatively related to the disposition effect.*


*P7:* 
*Risk preference mediates the negative relationship between the family wealth and disposition effect.*


### 2.4. Deposition Effect and Asset Return

There is a momentum effect in stock return; stocks that have performed well recently are more likely to perform well in the future, while those with poor recent performance are more likely to underperform [28]. Many other asset markets such as the currency market, real estate market, bond market, and future market have a similar effect [30,31,32]. Consequently, the disposition effect is harmful to asset returns. Empirical evidence supports this viewpoint [33,34]. Specifically speaking, Seru et al. (2010) find that the disposition effect has a negative impact on the performance of stock investors [33]. Choe and Eom (2009) find a negative relationship between the disposition effect and future investor performance [34]. Therefore, this research proposes the following:

*P8:* 
*Individuals whose investment behavior shows a higher disposition effect receive lower asset returns.*


*P9:* 
*The risk preference and disposition effect serially mediate the positive relationship between family wealth and asset return.*


### 2.5. The Moderating Role of Financial Literacy

Financial literacy refers to the mixture of awareness, attitude, knowledge, skill, and behavior that is necessary for investors to make sound financial decisions and achieve financial well-being ultimately [35]. Scholars usually measure financial literacy objectively based on respondents’ correct response rate to a financial knowledge questionnaire [36,37,38]. Empirical research has found that high financial literacy can decrease investment behavior biases, such as the disposition effect and herding bias [38]. There is also empirical evidence suggesting that financial literacy can lead to better financial decisions, such as optimal stock market timing, suitable stock selection, proper investment portfolio construction, and avoidance of harmful investment advice [37,39].

Additionally, a positive link has been reported between financial literacy and sound financial practices, such as cash flow management, credit management, and savings [8,10,36]. Although Graham et al. (2009) found that self-assessment investment competence is positively related to trading frequency bias [40], it is worth noting that self-assessment investment competence differs from financial literacy, as discussed in this paper. In academic studies, finance literacy is usually measured based on objective questions, not subjective ones. Therefore, the current literature shows that high financial literacy can significantly improve investment behavior and decisions [9].

This research has discussed the systematic impact of risk preference on the asset-holding periods and the disposition effect. Likewise, financial literacy emerges as a crucial psychological resource as it correlates with enhanced investment decisions and returns [37,39]. However, risk preference and financial literacy are two different types of resources. By definition, risk preference centers on the propensity to embark on risky endeavors, whereas financial literacy pertains more to investment ability. The performance function usually includes motivation and ability as independent variables [41]. Moreover, it is commonplace for the ability to offset partial deficiencies in motivation. Financial literacy can aid investors in recognizing and subsequently avoiding their own investment behavioral biases. It has been found that financial literacy can decrease investment behavior bias such as the disposition effect and herding bias [38]. Moreover, it is highly likely that financial literacy can increase investor’s holding period because high financial literacy can help investor awareness of the importance of the holding period and the harm of frequent transactions. As a result, it is reasonable that financial literacy can mitigate the negative impact of low-risk preference on investment behaviors, such as the asset-holding period and the disposition effect. Therefore, this paper proposes the following:

*P10:* 
*Financial literacy moderates the positive relationship between the risk preference and asset-holding period.*


*P10a:* 
*When financial literacy is high, the positive relationship between the risk preference and asset-holding period becomes weaker.*


*P10b:* 
*When financial literacy is low, the positive relationship between the risk preference and asset-holding period becomes stronger.*


*P11:* 
*Financial literacy moderates the negative relationship between risk preference and the disposition effect.*


*P11a:* 
*When financial literacy is high, the negative relationship between risk preference and the disposition effect becomes weaker.*


*P11b:* 
*When financial literacy is low, the negative relationship between risk preference and the disposition effect becomes stronger.*


## 3. Materials and Methods

This study uses a literature review and theoretical analysis as the main research methods. Through review of related mature theories and logical inference, this study proposes an investment psychology model (Figure 2) for explaining wealth inequality. In this process, the author explains the appropriateness of related theories for constructing the model and applies related theories to analyzing the specific relationships in the model. The main mature theories utilized by the author are the conservation of resources (COR) theory, prospect theory, and performance determinants theory. Specifically speaking, the COR theory provides an overall analysis framework for this paper. Prospect theory is used to explore the relationship between the risk preference and disposition effect. Moreover, the performance determinants theory is utilized to analyze the moderating role of financial literacy in the relationship between the risk preference, disposition effect, and asset-holding period.

Here are some reasons for the choice of research method. First, the research area of this paper (investment skill factors influencing wealth return heterogeneity) is relatively new and lacks a theoretical framework. This paper can provide theoretical guidance for subsequent empirical research. Second, it is difficult for ordinary researchers to collect related data for validating the proposed model. Collecting and matching the data about the financial literacy and disposition effect of investors usually requires researchers to have a very close cooperation relationship with corporations in the financial sector such as banks. Third, the main aim of this paper is the theoretical exploration of a novel research topic, rather than empirical validation.

## 4. Conclusions and Discussion

### 4.1. Summary of Proposed Model

The proposed model aims to partially explain the phenomenon whereby wealthy individuals receive higher returns from assets than less wealthy individuals, even when both groups hold the same type of asset. This phenomenon was identified by Fagereng (2020) and suggests that investment skill factors contribute to wealth return heterogeneity [6]. As a result, this research has identified two key investment skill factors—the asset-holding period and disposition effect—and has developed a model to incorporate these factors into the explanation for wealth return heterogeneity.

The overall theoretical framework of the proposed model is based on COR theory. The basic logic of the model revolves around the initial resource effect proposed by COR theory, which posits that individuals with rich initial resources show more positive resource investment behavior and stronger resource acquisition capabilities, thereby enabling them to amass more resources [15]. Specifically, wealthy individuals exhibit higher risk preferences, which is an important psychological resource. This psychological resource fosters positive resource investment behavior, such as longer asset-holding periods and a reduced disposition effect. These behaviors help wealthy individuals achieve more financial returns. Additionally, the proposed model suggests that another psychological resource, financial literacy, can compensate for deficiencies in risk preference. This concept is derived from performance determinants theory, which proposes that ability factors can partially offset the shortcomings of motivational factors. Specifically, financial literacy can attenuate the positive relationship between the risk preference and asset-holding period, as well as the negative relationship between risk preference and the disposition effect.

### 4.2. Theoretical Implication

Previously, the application of COR theory mainly concentrated on organizational psychology [15]. This paper expands the application of COR theory to the area of investment psychology. Specifically, it utilizes the initial resource effect proposed by COR theory to analyze the problem of wealth return heterogeneity. This not only offers a new theoretical framework for addressing this issue but also introduces a fresh theoretical perspective into investment psychology. Indeed, financial investment behavior is a typical resource investment behavior. It consumes investors’ cognitive, psychological, financial, and even social resources while being associated with further resource acquisition. As a result, COR theory proves to be a highly suitable framework for analyzing investment psychology and holds great potential to explain a broader array of financial investment phenomena. Moreover, this paper also contributes to expanding the scope of COR theory by integrating the concept of risk preference as a crucial psychological resource, an element traditionally excluded from the COR framework [15,17]. Given empirical findings in the financial market demonstrating that higher risk preference is associated with higher return (resource gaining) [5,6], it is reasonable for scholars to consider risk preference as a significant psychological resource.

In addition, this paper contributes to the wealth inequality literature by offering new theoretical insights into wealth return heterogeneity. A recent empirical finding suggests that investment skill factors contribute to wealth return heterogeneity [6]. However, research exploring the relationship between specific investment skill factors and wealth return heterogeneity is scarce. The current limited theoretical explanation places a spotlight on information availability, financial literacy, and information process ability [5,6,7], often overlooking the role of asset-holding periods in the explanation of wealth return heterogeneity.

In contrast, the model proposed in this research analyzes the serial mediating role of the risk preference and asset-holding period in the relationship between family wealth and asset return, thereby enriching the investment skill explanation of wealth return heterogeneity. This paper introduces a new perspective on investment skill explanation. Previous explanations, such as information availability, financial literacy, or information process ability, primarily address opportunity and ability factors that influence investment skills [5,6,7]. The two main variables in this research—the risk preference and asset-holding period—are more about motivational factors that influence investment skills. The underlying logic of this explanation is that individuals with high-risk preferences are more willing to hold assets for a longer period. This implies that other scholars aiming to improve the investment skill explanation could benefit from focusing more on motivational factors in their research.

Moreover, this paper contributes to the literature about the relationship between the wealth and disposition effect by constructing a new explanation for this relationship. Previous scholarly perspectives attributed this correlation to wealthy individuals’ access to professional financial advice and superior information processing abilities [7]. This paper introduces a new explanation by positing risk preference as a mediator in the relationship between wealth and the disposition effect. This explanation also utilizes the theoretical framework of prospect theory. Originally, prospect theory demonstrated the changes in the risk-taking behavior of the same individual in contrasting situations (gains and losses) [28]. This paper applies prospect theory to compare the risk-taking behavior of individuals with different levels of risk preference by analyzing their sensitivity to returns, which extends the application of prospect theory. The analysis shows that risk preference is negatively related to the disposition effect. This analysis result is consistent with the simulation results of Andrikogiannopoulou and Papakonstantinou (2020) [29]. A major difference between this research analysis and that of Andrikogiannopoulou and Papakonstantinou (2020) is that this paper considers individuals with low-risk preferences to be more sensitive to both losses and gains [29]. Therefore, this research constructs value functions that differ from those of Andrikogiannopoulou and Papakonstantinou (2020), thereby enriching prospect theory [29].

Additionally, this paper contributes to performance management theory by extending its application to the area of investment psychology. Traditional performance theory proposes that individual performance is contingent upon three factors: ability, motivation, and opportunity [41]. These three factors can complement each other [41]. Following this logic, the present study analyzes the moderating role of financial literacy in the nexus between the risk preference, asset-holding period, and disposition effect. This analysis not only extends the application of performance management theory but also enriches the literature on the relationship between the risk preference, asset-holding period, and disposition effect. This analysis implies that scholars can analyze wealth return heterogeneity within the framework of performance management theory and attribute the wealth return heterogeneity to the corresponding ability factor, motivation factor, or opportunity factor.

### 4.3. Practical Implication

The model proposed in this paper has important practical implications, as it reveals the investment psychology mechanism that contributes to the disadvantaged position of less wealthy people and identifies financial literacy as a key factor that can mitigate this mechanism. It is well documented that financial literacy levels are generally low worldwide, regardless of a country’s stage of economic development [9,38]. Therefore, this paper suggests that one effective method to decrease wealth inequality and improve the position of less wealthy individuals is to improve the financial literacy of citizens, especially those who are less affluent. Additionally, another way to decrease wealth inequality is to help less wealthy individuals avoid the negative effects of low financial literacy. Next, this study will discuss how to implement these two methods in practice.

To improve the financial literacy of citizens, it is necessary for governments to conduct a nationwide survey to assess the financial literacy situation of citizens [9]. Surveys can help policy-makers understand the overall levels of financial literacy and identify group differences, which facilitates targeted and effective interventions. According to the current literature, female investors are a disadvantaged group in financial literacy [42,43]. So, policy-makers should carefully consider the characteristics of female investors when proposing related interventions. Moreover, it is recommended that these surveys use objective measures of financial literacy because the results from objective assessments differ significantly from subjective assessments and more accurately reflect the actual level of financial literacy [9].

There is extensive empirical evidence supporting the significant positive impact of financial education on financial literacy [9,44,45]. Moreover, empirical research underscores the significant effectiveness of financial education for individuals with low income and education levels [44]. As a result, to mitigate wealth inequality, it is imperative for the government to provide ample opportunities for less wealthy individuals to receive financial education. Governments can incorporate financial courses into the compulsory education system and offer free financial education programs targeted at less wealthy citizens. Furthermore, to improve the effectiveness of financial education courses or programs, it is necessary to increase their intensity. For a 10-week course or program, a recommended intensity of 8 h of learning time per week is advisable [45]. In addition, a good financial education program should prepare high-quality and certificated teachers and link core financial concepts to challenges investors face in reality [46].

It has been proven that access to financial advisors can significantly benefit individual investors [47]. Scholars consider professional services from financial advisors as an effective way to complement the shortcomings in financial literacy among individual investors [9]. As a result, it is crucial to provide affordable access to financial advisors for less wealthy citizens, as they have fewer opportunities and less purchase power to obtain professional financial advice. To solve this problem, governments and charitable organizations should endeavor to provide low-cost financial advice services to less wealthy citizens. Additionally, governments can encourage profit-oriented management organizations to offer such services by providing corresponding tax incentives.

### 4.4. Limitations and Future Research

Investment skill factors contributing to wealth return heterogeneity represent a new theoretical perspective that has recently gained empirical support in wealth inequality research [4,6]. This paper explores the topic by integrating a novel theoretical framework, COR theory, into investment psychology, highlighting the role of two investment skill factors: the asset-holding period and disposition effect. However, it is highly likely that there are additional investment skill factors contributing to wealth return heterogeneity. In the future, researchers can utilize COR theory or other suitable frameworks to explore these factors.

Within the COR theory framework, several research directions warrant further investigation. A key mechanism discussed in this paper is risk preference, an important psychological resource that can improve resource investment skills, such as a longer asset-holding period and reduced disposition effect. Future studies should explore another psychological, cognitive, or social resource of wealthy individuals that can lead to enhanced investment skills. It is important to note that better investment skills usually entail the avoidance of behavioral biases. Therefore, future studies could explore related investment skill factors from the perspective of behavioral bias avoidance.

Moreover, this research utilizes the theory about determinants of performance to enrich the explanation of investment skills for wealth return heterogeneity. The related theory proposes that the determinants of performance mainly include the ability factor, motivation factor, and opportunity factors, all of which interact synergistically [41]. Within this framework, this paper analyses the impact of one type of motivation factor (risk preference) and one type of ability factor (financial literacy) on investment behaviors, specifically the asset-holding period and the deposition effect. Future scholars can leverage this framework to further investigate other motivation factors, ability factors, and opportunity factors that contribute to the disparities in investment skills among individuals across different wealth levels.

In addition, one common limitation of theoretical research lies in the potential requirement for a further empirical validation of its findings. The validity of the research findings of this research should be confirmed through future empirical research. As a result, this research offers some advice for scholars who are interested in validating the findings of this research. As highlighted in the Introduction, while collecting the necessary data to validate the model proposed in this paper is challenging, it remains feasible. Scholars can use the data collection method used in the paper by Guiso and Viviano (2015) [37]. In that research, the scholars collaborated closely with a bank to collect individual transaction data from the bank’s database and gathered data about financial literacy by asking the bank to survey individual customers on it. Then, they matched two types of data together with the help of the bank. This data collection method usually requires scholars to establish very deep cooperation and trust with the bank.

Moreover, this study also provides guidance for measuring the main variables in the model. For the measurement of the family wealth, disposition effect, and financial literacy, scholars can draw from established methods from the related literature such as those of Guiso and Viviano (2015) and Dhar and Zhu (2009) [7,37]. Notably, to measure the asset-holding period of a certain type of asset, it is recommended to use a value-weighted average holding period of all single assets of the same type. This measure of the asset-holding period correlates more strongly with risk preference. As for the measure of risk preference, an investment risk preference scale is preferable over a general risk preference scale. Lastly, the asset return of a certain type should be measured as the total asset return of all single assets of the same type at the endpoint of a certain time period.

## Figures and Tables

**Figure 1 behavsci-14-01048-f001:**
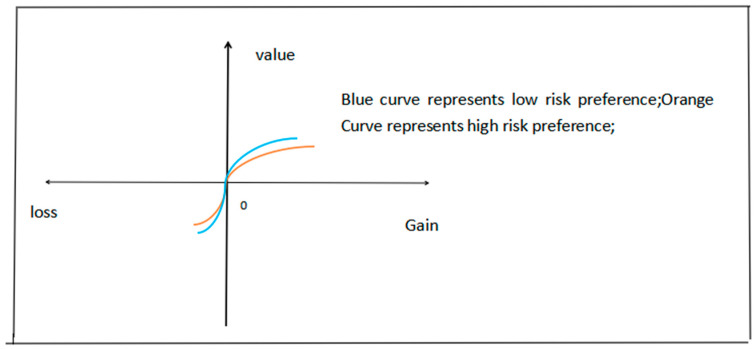
Comparison of value curve.

**Figure 2 behavsci-14-01048-f002:**
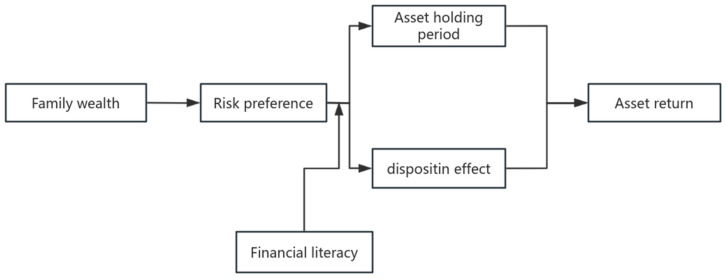
The proposed model.

## Data Availability

The author did not analyze or generate any datasets, because the work proceeded within a theoretical and mathematical approach.

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
