# Peer review of "The Impact of Family Wealth on Asset Return: A Moderated Chain Median Model Partially Explaining Wealth Inequality"

_behavsci, 2024, doi:10.3390/bs14111048_

Round 1
Reviewer 1 Report (Previous Reviewer 2)
Comments and Suggestions for Authors
I think the paper has improved in many aspects. However, based on the proposed models, the results obtained by the author are not clear to me. I think a further explanation and justification would be necessary.
Author Response
Dear reviwer:
Thank you for your comments. Please see my point-to-point response in the attachment.

Reviewer 2 Report (New Reviewer)
Comments and Suggestions for Authors
MAJOR COMMENTS
The research topic is relatively new. The author aims to explore the wealth inequality phenomenon from the perspective of investment psychology.
The article can provide good information.
In the Literature Review section the readers can find the necessary background for the present research.
However, I think the paper needs a reorganization. The article should comprise the section Materials and Methods in order to make the methodology clear. This section is found in Literature review section.
MINOR COMMENTS
The author is referred to as `this writer'. I don't think such an address is appropriate.
Author Response
Dear reviewer:
Thank you for your comments. Please see my point-to-point response in the attachment.

Round 2
Reviewer 2 Report (New Reviewer)
Comments and Suggestions for Authors
Thank you for considering my comments and revising the manuscript accordingly
This manuscript is a resubmission of an earlier submission. The following is a list of the peer review reports and author responses from that submission.
Round 1
Reviewer 1 Report
Comments and Suggestions for Authors
In this paper, the author presents a psychological model of asset returns. In this model, the returns on an asset depend on the disposition effect and the asset holding period, factors that are determined by family wealth, risk preferences, and financial literacy.
While I’m sympathetic to the idea of applying COR theory to understand heterogeneity in asset returns, the paper presents major theoretical and conceptual flaws.
As an example of a major theoretical flaw, in lines 108 to 115, the author writes:
“Risk preference is an important psychological resource because it is positively associated with investment returns (wealth) (Bach et al., 2019; Fagereng et al., 2020). According to COR theory, individuals with more resources are more capable of gaining additional resources compared to those with fewer resources (Hobfoll et al., 2018). As a result, based on COR theory, it can be inferred that individuals with greater family wealth exhibit higher risk preferences. This viewpoint is also supported by empirical studies (Bach et al., 2019; Fagereng et al., 2020).”
That is:
Higher risk preferences are associated with higher investment returns, so they are a psychological resource.
Resources bring more resources (COR theory)
So family wealth (resource) implies a higher risk preference (resource).
Using a broad definition of what a resource is, this logic could be used to justify almost any relationship. For example, one could use it to justify that having a higher family wealth (resource) will make you taller (resource). This example illustrates the lack of logical reasoning and the extent to which the arguments employed in the theoretical argumentation are limited.
Conceptually, the author needs to revise the economics literature on risk preferences (which is cited, but not properly understood in many cases). For example, the author cites Mata et al. (2008) when defining risk preferences (line 108)
“Risk preference refers to the degree to which an individual prefers to engage in ac-
tivities or behaviors that exhibit significant variation in returns, irrespective of whether they result in losses or gains (Meta et al., 2018).”
In this definition, the authors remarks that risk preferences are irrespective of whether an investment results in gains or losses. But one of the key conceptual parts of the paper links risk preferences to the disposition effect (the tendency to hold losing assets and sell winning ones….). Again, this is just an example of a major conceptual flaw (there are others).
Reviewer 2 Report
Comments and Suggestions for Authors
The topic of the paper "A theoretical model of investment psychology partially explaining wealth inequality problems" seems interesting to me.
I am going to make some comments about it
The introduction does not seem clear to me. I think that the objectives of the paper are not clearly specified. I think that several aspects are mixed (the role of risk, inequality, financial literacy)
The title of the paper does not seem to me to reflect its content.
The introduction should more clearly state the objectives of the paper, motivate its importance and present its structure.
In section 2 Literature review and propositions it begins with 2.1. Family wealth, risk preference, asset holding period and asset return: conservation of resources (COR) theory perspective. I think that some sentences explaining the meaning of section 2.1 would be necessary. On the other hand, this title seems quite ambiguous to me.
I think that a further explanation is needed about the function v(x) and the parameters a and b (line 186) and v(x) and the parameters g and r (line 200)
From the proposed models, the results obtained by the author are not clear to me, I think that a further explanation and justification would be necessary
The numbering of section 2 is not correct, it needs to be corrected
Separate subsection 3.4 from section 3
On the other hand, 12 propositions seem excessive to me for a single paper, I think it is excessively ambitious
Reviewer 3 Report
Comments and Suggestions for Authors
Dear Author,
Thank you for submitting your manuscript. While I find your proposal has potential, I believe it would greatly benefit from a deeper exploration of the background and a clearer articulation of its contributions to the existing literature. Specifically, there appears to be a lack of well-referenced sources on the topic, which is essential for establishing a solid foundation for your work. I encourage you to conduct a thorough analysis of relevant literature and incorporate it into your manuscript, emphasizing how your research adds to the current understanding of the subject. I look forward to reviewing a revised version that addresses these points in greater depth.
Best regards
The reviewer
Aristei, D., & Gallo, M. (2022). Assessing gender gaps in financial knowledge and self-confidence: Evidence from international data. Finance Research Letters, 46, 102200.
Behrman, J. R., Mitchell, O. S., Soo, C. K., & Bravo, D. (2012). How financial literacy affects household wealth accumulation. American Economic Review, 102(3), 300-304.
Campbell, J.Y. (2006). Household finance. The Journal of Finance, 61(4), 1553-1604.
Cupák, A., Fessler, P., & Schneebaum, A. (2021). Gender differences in risky asset behavior: The importance of self-confidence and financial literacy. Finance Research Letters, 42, 101880.
Hira, T.K. (2012). Promoting sustainable financial behaviour: Implications for education and research. International Journal of Consumer Studies, 36(5), 502-507.
Ingale, K.K., & Paluri, R.A. (2020). Financial literacy and financial behaviour: A bibliometric analysis. Review of Behavioral Finance.
Lusardi, A., & Mitchell, O. S. (2014). The economic importance of financial literacy: Theory and evidence. Journal of Economic Literature, 52(1), 5-44.
Nave, J. M., Oliva, L., & Toscano, D. (2023). Financial knowledge and financial behaviour: The moderating role of home ownership. Finance Research Letters, 57, 104208.
OECD (2018). OECD/INFE Toolkit for Measuring Financial Literacy and Financial Inclusion. OECD, Paris.
Robb, C. A., & Woodyard, A. (2011). Financial knowledge and best practice behavior. Journal of Financial Counseling and Planning, 22(1).
Sekita, S., Kakkar, V., & Ogaki, M. (2022). Wealth, Financial Literacy and Behavioral Biases in Japan: the Effects of Various Types of Financial Literacy. Journal of the Japanese and International Economies, 64, 101190.
Van Rooij, M., Lusardi, A., & Alessie, R. (2011). Financial literacy and stock market participation. Journal of Financial Economics, 101(2), 449-472.
Vestman, R. (2019). Limited stock market participation among renters and homeowners. The Review of Financial Studies, 32(4), 1494-1535.
Yamori, N., & Ueyama, H. (2022). Financial Literacy and Low Stock Market Participation of Japanese Households. Finance Research Letters, 44, 102074.
